# Segmental Evaluation of Thoracic Aortic Calcium and Their Relations with Cardiovascular Risk Factors in the Brazilian Longitudinal Study of Adult Health (ELSA-Brasil)

**DOI:** 10.3390/cells10051243

**Published:** 2021-05-18

**Authors:** Jesiana Ferreira Pedrosa, Luisa Campos Caldeira Brant, Stephanie Alves de Aquino, Antonio Luiz Ribeiro, Sandhi Maria Barreto

**Affiliations:** 1Department of Image & Anatomy, Faculty of Medicine, Universidade Federal de Minas Gerais, Belo Horizonte 30130-100, Brazil; jesianafp@gmail.com; 2Departament of Internal Medicine, Faculty of Medicine & Clinical Hospital, Universidade Federal de Minas Gerais, Belo Horizonte 30130-100, Brazil; luisabrant@gmail.com (L.C.C.B.); alpr1963br@gmail.com (A.L.R.); 3Longitudinal Study of Adult Health, Research Centre of Minas Gerais, Belo Horizonte 30130-100, Brazil; stephanieaquino.rad@gmail.com; 4Departament of Social and Preventive Medicine, Faculty of Medicine & Clinical Hospital, Universidade Federal de Minas Gerais, Belo Horizonte 30130-100, Minas Gerais, Brazil

**Keywords:** thoracic aorta, atherosclerosis, vascular calcification, risk factors

## Abstract

Thoracic aortic calcium (TAC) appears to be a subclinical marker of cardiovascular disease (CVD) and to predict cardiovascular (CV) mortality. However, studies on TAC use tomographic scans obtained for coronary artery calcium (CAC) score, which does not include the aortic arch. This study evaluates TAC prevalence in aortic arch (AAC), ascending (ATAC) and descending thoracic aorta (DTAC) and verify whether they are associated with the same CV risk factors. Cross-sectional analysis, including 2427 participants (mean age 55.6 ± 8.7; 54.1% women) of the ELSA-Brasil cohort. Nonenhanced ECG-gated tomographies were performed in 2015–2016. Multivariable logistic regression estimated the CV risk factors associated with calcium in each segment. Overall prevalence of ATAC, AAC and DTAC was, 23.1%, 62.1%, and 31.2%, respectively. About 90.4% of the individuals with TAC had AAC and only 19.5% had calcium in all segments. In the multivariable analysis, increasing age, lower levels of schooling, current smoking, higher body mass index, and hypertension remained associated with calcium in all segments. No sex or race/ethnicity differences were found in any aortic segment. Diabetes and dyslipidemia were associated with ATAC and DTAC, but not with AAC, suggesting that AAC may reflect an overlap of mechanisms that impact vascular health, including atherosclerosis.

## 1. Introduction

Thoracic aortic calcium (TAC) is a common imaging finding that reflects systemic atherosclerosis [1]. Besides, aortic wall calcium exacerbates arterial stiffening [2], which is associated with end-organ damage [3]. Several studies have shown that TAC and coronary artery calcium (CAC) are associated with distinct risk factors [4,5,6,7]. However, TAC has generally been analyzed using the same tomographic scan obtained for CAC score, which does not include the aortic arch, therefore studies that evaluated the three aortic segments separately are very scarce [8,9,10,11,12]. The calcium distribution along the aorta is typically heterogeneous [13]. Each aortic segment has different embryonic origin and is subject to diverse hemodynamic stress, which also appears to affect susceptibility to calcium deposition [14], once the rate of calcium differs among individuals [15]. Moreover, calcium in each aortic segment appears to have distinct predictive values for CV and non-CV morbidity and mortality [9,16,17,18,19], and may differ in the associated CV risk factors [20]. Thus, the present study aimed to evaluate the prevalence of TAC by segments (aortic arch, descending and ascending aorta) and to verify if they differ regarding the kind and the magnitudes of the associations with traditional cardiovascular (CV) risk factors in adult participants without overt CVD.

## 2. Methods

### 2.1. Study Population

This study is part of the Longitudinal Study of Adult Health (ELSA-Brasil), a multicenter cohort designed to investigate the determinants of CV disease and diabetes [21]. This prospective study was initiated in 2008 and included 15,105 civil servants, aged 35 to 74 years, active and retired employees of six Brazilian institutions [22]. The present study was conducted in the Minas Gerais Investigation Center in 2015–2016, after the second examination of ELSA-Brasil cohort, which enrolled 2923 participants. Pregnancy, postpartum, breast-feeding (until 6 months post childbirth), exposure to radiation at work, any piece of metal in the chest, current radiotherapy, nonparticipation in second visit to Investigation Center and refusal to perform the exam were defined as exclusion criteria. Multislice computed tomography (MSCT) was performed in 2638 participants. Individuals that reported medical history of myocardial infarction (*n* = 15), congestive heart failure (*n* = 21), stroke (*n* = 24) and/or cardiac surgery (*n* = 13) were excluded from the analysis. Additionally, 142 participants were submitted to a different scan protocol, in which the aortic arch was not included. Thus, 2427 participants were included in the present study.

The study protocol has been priorly approved by the Universidade Federal de Minas Gerais’s Ethics Committee on research in human, approval number 47125015.4.1001.5149, and our protocol conforms to the Ethical Guidelines of the 1975 Declaration of Helsinki. All participants signed a written informed consent.

### 2.2. Cardiovascular Risk Assessment

Medical history, anthropometric measurements, and laboratorial data for the present study were obtained during visit 2 of the ELSA-Brasil study (2012–2014), by face-to-face interviews and exams using standardized protocols. Race/skin color were self-reported and categorized according to Brazilian Census into white; brown or “pardo”; black; and others (Indigenous and Asian were combined due to small numbers) [22]. Educational level was classified as university degree; complete secondary school; complete elementary school; and incomplete elementary school. Smoking was defined as never, former, and current smoker. The “smoker groups” (former and current) included individuals who had smoked at least 100 cigarettes (5 packs) throughout their life, with those who did not smoke in the last 30 days considered as “former smoker” [23]. Physical activity was classified as insufficient, moderate, and vigorous based on the long version of the International Physical Activity Questionnaire [24]. Excessive use of alcohol was defined as ≥210 g of ethanol per week for men and ≥140 g for women [22]. Family history of cardiovascular disease (CVD) was defined as the report of acute myocardial infarction, myocardial revascularization or sudden death in first-degree relatives (men aged ≤55 and women ≤ 65 years old) [21]. Body mass index (BMI) was obtained by weight (Kg)/height squared (m^2^). Body Surface (BS) was estimated by Dubois & Dubois method as BS = 0.007184 × height^0.725^ × weight^0.425^). Resting blood pressure was measured three times in the seated position, in the left arm, 2 cm above the cubital fossa, using an Omron model 765CP automated oscillometric sphygmomanometer (Omron, Kyoto, Japan) and the average of the second and third readings of systolic and diastolic blood pressure were recorded and treated as continuous variables (mmHg) [25]. Blood samples were collected after 12 h overnight fast to determine glucose, glycohemoglobin, total cholesterol, high-density lipoprotein (HDL)-cholesterol and low-density lipoprotein (LDL)-cholesterol levels [21]. The use of continuous medication in the previous 2 weeks were ascertained by interview, and visualization of medical prescriptions, medicine packaging and blisters and the use of antidiabetic, anti-hypertensive, and lipid lowering medications identified [26]. Hypertension was defined by systolic blood pressure (SBP) ≥ 140 mmHg and/or diastolic blood pressure (DBP) ≥ 90 mmHg and/or use of antihypertensive drugs. Diabetes mellitus (DM) was defined by fasting glucose ≥ 126 mg/dL, postprandial glucose ≥ 200 mg/dL, or glycohemoglobin ≥ 6.5%, in addition to “being treated for DM” or reported medical diagnosis of DM. Dyslipidemia was defined by total cholesterol ≥ 240 mg/dL, LDL-cholesterol ≥ 160 mg/dL, HDL-cholesterol ≤ 40 mg/dL or use of lipid lowering medications.

### 2.3. Imaging Techniques

All participants underwent the same 64-slice MSCT scanner (Lightspeed, General Electric, Chicago, IL, USA). The scanogram encompassed from 1 cm above the top of the aortic arch to the heart apex. The method has been reported previously [4]. The CT scan parameters were 2.5 mm thick slices with 20 × 0.62 mm collimation, 120 kVp, 100 mAs and prospective ECG triggering at 70% of the cardiac cycle. The reconstruction algorithm used body filter. The media of effective dose calculated was 1.75 mSv.

The images were firstly analyzed by an experienced radiologist to identify the presence of calcium in the thoracic aorta (TAC) as a whole. Secondly, for the present study, all the images were reviewed by the radiologist together with a qualified technologist to define the segments that calcium was present. Finally, an inter- and intra-observer correlation study was performed with a random sample of 50 CT scans, which were scored twice by the radiologist and once by a second radiologist with 10 years of experience, resulting in intraclass correlation coefficients higher than 0.99 for intra and interobserver analysis.

Calcium was identified using semiautomatic software (Smart Score v4.0), that highlighted in green all calcium based on a threshold of 130 Hounsfield Unit (HU) and calculated the Agatston score [27]. The observer went over every axial image and delineated the calcium localized in the arterial beds to be validated. The anatomical references to evaluate the thoracic aorta by segments have been previously described [13]. In summary, the ascending thoracic aortic calcium (ATAC) was considered from the sinotubular junction until the lower edge of pulmonary artery bifurcation, therefore calcium from Valsalva sinus and aortic valve was not included. The descending thoracic aortic calcium (DTAC) was defined from the level of the lower edge of pulmonary artery bifurcation to the heart apex. Consequently, the aortic arch calcium (AAC) was found above ATAC and DTAC using the same anatomical level as reference (the lower edge of pulmonary artery bifurcation).

### 2.4. Statistical Analyses

Descriptive statistics were presented as mean and standard deviation for continuous variables and frequency distribution for categorical variables. Continuous variables were compared with Student *t* test and categorical variable with Pearson’s chi-square test. The prevalences of ATAC, AAC and DTAC were dichotomized as present (Agatston > 0) or absent (= 0). The prevalence of calcium in each segment, plus the ascending and descending segments together (ADTAC) was presented according to sociodemographic, lifestyle, and clinical factors. The associations of these factors with calcium in each segment separately were assessed using multivariable logistic regression analysis. After the univariable analysis (model 0), sociodemographic factors were added (Model 1). Subsequently, we included the lifestyle factors (smoking, physical activity, and excessive use of alcohol), BMI, and family history of premature CVD (Model 2); and finally, we included the remaining risk factors (hypertension, dyslipidemia and diabetes) (Model 3). All associated covariables with *p* < 0.20 in the univariable analysis were considered in the multivariable models, but only the variables that remained statistically associated at the level of *p* < 0.05 remained in the final analysis (Model 3). The confidence interval corresponded to 95%. Statistical analysis was performed with Stata/MP v14.0 for MAC (StataCorp LP, College Station, TX, USA).

## 3. Results

### 3.1. Patient Characteristics

The clinical characteristics of the study population is shown in Table 1. Overall, the mean age was 55.6 ± 8.7 years, 54% women, nearly half were whites (49%), and the majority had university-level education (67%). Considering the clinical conditions, 38.9% had hypertension, 41.2% had dyslipidemia, and 16.3% were diabetic.

### 3.2. Calcium Distribution along Thoracic Aorta

Overall, 1669 (68.8%) participants demonstrated TAC. A total of 1508 (62.1%) individuals had AAC, which corresponds to 90.4% of the population with TAC. The prevalence of ATAC was 23.1%, DTAC prevalence was 31.2% and the combination of ATAC and DTAC showed a prevalence of 40.3%. About 13.4% of TAC population had calcium in all segments and 28.4% had only AAC. Figure 1 shows a Venn diagram with the intersections of the study participants according to the distribution of TAC by segments. Figure 2 demonstrates no sex-related differences in the prevalence of calcium in any aortic segments. Table 1 also show the prevalence of calcium in each segment according to selected risk factors.

### 3.3. Association between Risk Factors and Calcification

The results of the multivariable analysis for CV risk factors with the presence of ATAC, AAC and DTAC are shown in Table 2.

For all thoracic aortic segments, higher age and BMI were associated with presence of calcium. For instance, an increase in one year of age raises in 10% the odds of AAC > 0. Similarly, any addition of one kg/m^2^ in the BMI would the chances of AAC by AAC > 0 in 10%. Of the variables included in Table 1, there were no sex-related or race/skin color differences in any TAC segment in the multivariable analysis. Lower levels of schooling, current smoking, hypertension, diabetes and dyslipidemia remained associated with ATAC and DTAC, however DTAC was more strongly associated with hypertension and ATAC with smoking and metabolic risk factors. As shown in Table 2, the factors associated with ADTAC agree with most of the factors associated with DTAC than with the other segments, except for family history of CVD, probably because of the prevalence of calcification is highest in the descending aorta. The relation of AAC to risk factors is similar to the above, except for the lack of association with diabetes and dyslipidemia and a positive association with the presence of a family history of CVD. The results were about the same when we run the models including diastolic and systolic blood pressure and antihypertensive medication use instead of hypertension (Appendix A). We also added body surface to the final models and the results were basically unchanged. In any of the models, body surface was statistically associated with TAC in the aortic segments (Appendix A).

## 4. Discussion

In this population-based cohort of adults, the prevalence of arterial calcification in the aorta was two to three-fold higher in the aortic arch comparing to the ascending and descending aorta. Most of the cardiovascular risk factors associated to the three studied segments were similar and included increasing age, lower levels of schooling, current smoking, higher BMI, and hypertension. Diabetes and dyslipidemia, metabolic risk factors, strongly related to coronary artery disease events, were associated with ATAC and DTAC, but not with AAC. Moreover, there was no difference in the prevalence of calcifications in any aortic segments related to sex, differing from what is known for CAC, for which men have higher burden of calcifications. So, despite the higher prevalence of AAC compared to other segments, the extended CT acquisition to include the aortic arch in the CAC studies seems to add little information regarding to the associations with cardiovascular risk factors.

Calcification distribution differs among TAC segments. We found the highest calcium concentration in the aortic arch (62%) and the lowest in the ascending aorta (23%). Few studies have compared prevalences of TAC by segments. Craiem et al. studied the segments of thoracic aorta, separately, and found results similar to ours. While the authors showed 64% of total TAC prevalence and 31% of prevalence for AAC only, our results were 62.1% and 28.4%, respectively [28]. Although, comparisons with other studies should be made with caution, because of differences associated with the population risk profile, AAC seems to consistently concentrate around 50% of the calcium of the thoracic aorta. Finally, in the present study the thoracic aorta segmentation did not follow the anatomical landmarks, which traditionally consider the aortic arch the segment between the origin of the brachiocephalic trunk and the origin of the left subclavian artery, once our aim was to evaluate the inclusion of aortic arch in the CAC studies. Probably, this division may have underestimated ATAC and DTAC prevalence and overestimated the aortic arch prevalence. However, we must stress that our anatomical definitions coincide with those presented in most of the cohort studies that evaluated ATAC and DTAC.

The ATAC prevalence vary quite widely among studies. The ATAC prevalence in ELSA (23.1%) was much lower than reported by the Heinz Nixdorf Recall Study (42.9%), but much greater than that found in the MESA cohort (5%). Such differences are not easily explained by variation in the study population characteristics. For instance, the Heinz Nixdorf Recall cohort is very similar to ours regarding age, sex, BMI and antihypertensive use, but has more smokers (23.1% vs. 9.5%), and less diabetics (7.5% vs. 16.2%) than ELSA. On the other hand, MESA cohort is slightly older (61.8 vs. 55.6 years), has less diabetics (12.5% vs. 16.2%) and more smokers (12.9% vs. 9.5) than our population. Moreover, the thoracic aorta segmentation was the same in the three cohorts, thus an unlike explanation for the wide variation in ATAC prevalence.

The differences in the prevalence of calcium in each aortic segment may be explained in part by the effect of sheer forces on the wall, once the ascending aorta usually is larger, has high blood velocity and no branches [29]. Though high velocity can cause shear stress, most atherosclerotic lesions occur at sites where shear stresses are low but rapidly fluctuating, such as branch vessels or where there are abrupt changes in vessel diameter, both mostly occurring in the aortic arch followed by the descending aortic segment [30]. The hemodynamic conditions with the vasculature mean that vascular smooth muscle cells (VSMCs) are under constant mechanical stress [31]. These mechanical forces, such as shear stress, have been linked to premature cellular aging, including oxidative stress leading to vascular calcification [31]. Further, the VSMCs colonizing the aortic arch derive from cardiac neural crest cells, whereas those populating the descending aorta derive from the mesoderm [14]. The VSMCs are central players in vessel wall inflammation and evidence has shown that VSMCs impact every step of atherosclerosis. Studies have demonstrated that the inflammatory process may critically depend on VSMCs plasticity and their ability to switch between different phenotypes [32]. One of the reasons that smooth muscle cells identification and fate tracing is complex is related to the differences in the embryonic origin of VSMCs [33].

Moreover, there are two mechanisms for aortic wall calcification: intimal (atherosclerotic) and medial (non-atherosclerotic). Advanced atheromatous lesions usually have arterial intimal calcification, while calcified and degenerated elastin and vascular smooth muscle cells have medial calcification and are frequently seen in patients with advanced age and chronic renal disease [34]. It is unclear if the latter pathophysiologic process of calcium formation in the aortic wall is a consequence of traditional risk factors or other local and systemic factors, and although both mechanisms may coexist in the same individual, they may differentially affect each aortic segment, influencing prevalence and association to specific risk factors. Unfortunately, CT scans cannot accurately differentiate intimal from medial calcifications, what would bring further insights to our findings.

Increasing age, lower levels of schooling, current smoking, higher BMI, and hypertension remained associated with all thoracic aortic segments in the final multivariable analysis. There is robust evidence that smoking, hypertension and body mass index are important risk factors for aortic calcification independently of the segment and even studying abdominal aorta [5,29,35,36,37]. Regarding smoking, the higher magnitude of association found with ATAC has been previously described and may be due to the nicotine effects in the cardiac function: raising blood pressure, heart rate, myocardial excitability and contractility—variables that more directly impact the ascending aorta [38]. A similar pattern of association was also found by Takasu et al. in the MESA cohort [29].

Differently, metabolic risk factors—diabetes and dyslipidemia—were not related to AAC in the final model, but were related to ATAC and DTAC. Our results may be reflecting a smaller influence of diabetes on aortic calcium deposition. Alternatively, the predominant mechanism of calcification in the AAC may be non-atheromatous, which could be less related to metabolic risk factors.

There were no sex-based differences in the prevalence of calcium in any aortic segment. Comparing to CAC, which are significantly more frequent in men [4], the absence of differences between sexes in all aortic segments reinforces the distinct pathologic processes involved in the calcium formation in different vascular beds [39]. However, this finding is controversial in previous studies. Takasu et al. investigated 6814 participants (49% men, mean age: 63 ± 10 years) and demonstrated higher prevalence of DTAC in women, while for ATAC it was only found in persons younger than 55 years [29]. In contrast, the Heinz Nixdorf Study evaluated 4025 individuals considering TAC as ATAC plus DTAC (47% men; mean age: 59 ± 9 years) and showed higher TAC prevalence in men [5]. In the small number of studies that included the aortic arch, women have been shown to concentrate more calcium in this segment than men [28,40,41]. As such, more studies are needed to elucidate this issue, particularly to better understand if TAC could be a more useful tool for risk stratification in women.

Importantly, any potential benefit in adding AAC evaluation to better estimate cardiovascular risk will need to be weighed against the extra radiation that individuals will have to be exposed comparing to CAC scans, though the radiation dose values in the present study did not exceed the “American Heart Association” recommendations for CAC screening [42].

The limitations of our study are related to the cross-sectional analysis that cannot establish the temporality of the associations. Since ELSA-Brasil is a cohort of civil servants in urban areas and our population came from one Brazilian state, the prevalence found cannot be extrapolated to the Brazilian population. On the other hand, the advantages of our study are a large multiethnic well-characterized sample; comprehensive measurements of calcifications in the segments of the thoracic aorta, separately, for which there is limited literature; and the adjustments for most risk factors for cardiovascular diseases.

## 5. Conclusions

In conclusion, we found greater prevalence of calcium in the aortic arch, comparing to the ascending and descending aorta and small differences in the associations between cardiovascular risk factors and the aortic segments, particularly due to the lack of association between diabetes and dyslipidemia to AAC. Because TAC has been shown to predict cardiovascular and non-cardiovascular mortality, our findings awaken the question about the importance of including the aortic arch in TAC analysis, which may reflect diverse mechanisms of calcium formation in the pathway to vascular disease. Future longitudinal studies evaluating the predictive value of calcium in each aortic segment are fundamental to clarify the importance of studying all of them as subclinical markers of morbidity and mortality.

## Figures and Tables

**Figure 1 cells-10-01243-f001:**
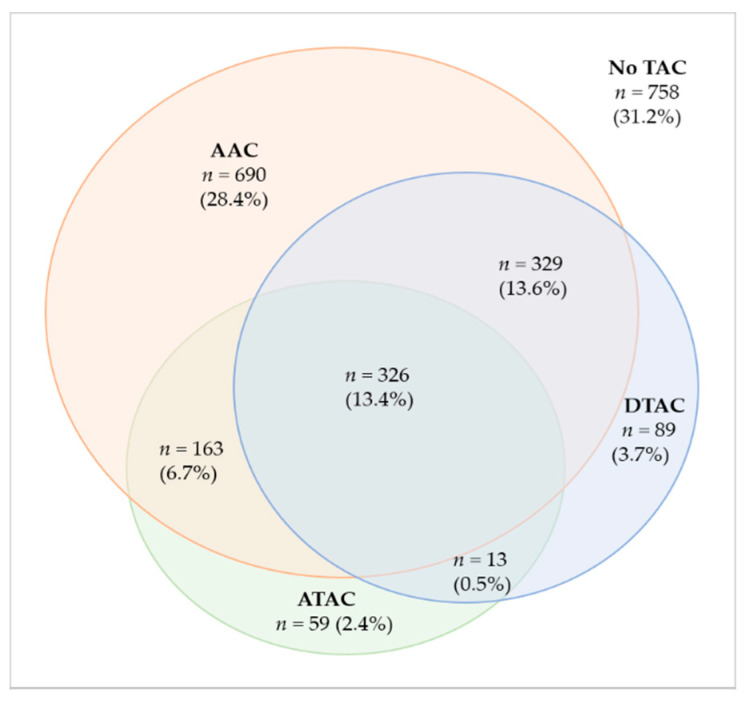
Venn diagram showing the distribution of calcium and the intersections of individuals according to aortic segments. The percentages refer to the total population (*n* = 2427).

**Figure 2 cells-10-01243-f002:**
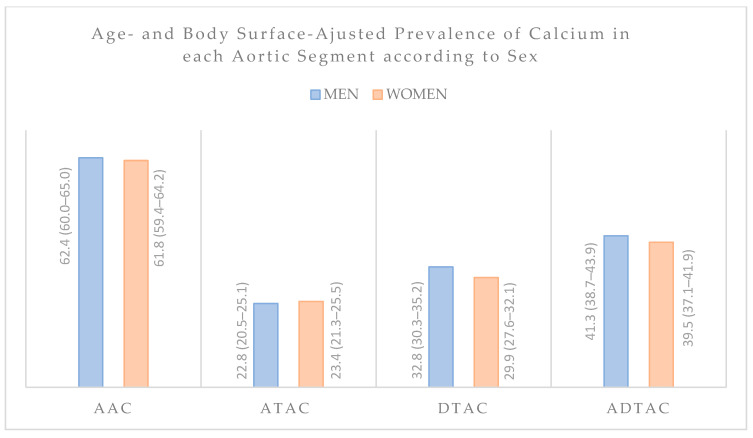
Clustered column graph demonstrating age- and body surface-adjusted prevalences of calcium in men and women in all aortic segments. AAC = aortic arch calcium; ATAC = ascending thoracic aortic calcium; DTAC = descending thoracic aortic calcium; ADTAC = ascending and descending thoracic aortic calcium.

**Table 1 cells-10-01243-t001:** Characteristics of study population and prevalence of calcium in the aortic arch, ascending aorta, descending thoracic aorta and in combination of calcium in the ascending and descending thoracic aortic segments (ELSA-Brasil, 2015–2016).

Variables	Population *n* = 2427 (100%)	AAC > 0 *n* = 1508 (62.1%)	ATAC > 0 *n* = 561 (23.1%)	DTAC > 0 *n* = 757 (31.2%)	ADTAC > 0 *n* = 979 (40.3%)
Age, years	55.59 ± 8.68	58.46 ± 8.25	61.07 ± 7.90	60.79 ± 8.50	60.13 ± 8.36
Women	54.10	53.85	54.72	51.78	53.01
Race/Skin color					
White	48.69	50.23	53.79	51.94	51.50
Brown	35.29	34.21	32.31	32.80	33.40
Black	13.31	13.21	11.37	12.99	12.31
Others (Asian, Indigenous)	2.71	2.35	2.53	2.28	2.79
Educational level					
University degree	67.01	63.57	63.21	62.70	62.99
Complete secondary	25.24	26.28	24.82	25.53	25.97
Complete elementary	4.21	5.57	5.71	5.56	5.11
Incomplete elementary	3.55	4.58	6.25	6.22	5.93
Smoker					
Never	61.48	55.11	45.63	54.03	53.01
Past	28.99	33.29	41.00	33.42	34.83
Current	9.53	11.60	13.37	12.55	12.16
Physical activity					
Insufficient	71.26	70.09	67.02	71.07	70.38
Moderate	20.45	22.35	25.13	20.87	21.65
Vigorous	8.29	7.56	7.84	8.06	7.97
Excessive use of alcohol	10.47	11.07	13.73	12.15	12.36
Family history of CVD	33.13	36.74	39.04	36.72	37.08
Body mass index, kg/m^2^	26.98 ± 4.73	27.44 ± 4.86	28.10 ± 5.30	27.75 ± 5.00	27.72 ± 5.01
Dyslipidemia	41.28	46.20	56.89	51.26	51.75
Hypertension	38.87	47.15	57.93	58.92	55.46
Diabetes	16.20	20.11	27.86	25.93	25.05

Values are percentages for categorical variables, means with standard deviation for variables with a normal distribution. AAC indicates aortic arch calcium; ATAC, ascending thoracic aortic calcium; CVD, cardiovascular disease; DTAC, descending thoracic aortic calcium; ADTAC, ascending plus descending thoracic aortic calcium.

**Table 2 cells-10-01243-t002:** Cardiovascular risk factors associated with the presence of calcium in the aortic arch, ascending aorta, descending thoracic aorta and in combination of calcium in the ascending and descending thoracic aortic segments in the final multivariable analysis. ELSA-Brasil, 2015–2016.

Variables	AAC > 0 OR (95% Cl)	ATAC > 0 OR (95% CI)	DTAC > 0 OR (95% CI)	ADTAC > 0 OR (95% CI)
Age (years)	1.12 (1.10–1.13) ‡	1.09 (1.08–1.11) ‡	1.11 (1.10–1.13) ‡	1.11 (1.10–1.13) ‡
Women	0.95 (0.78–1.16)	1.14 (0.91–1.41)	0.84 (0.68–1.02)	0.93 (0.77–1.13)
Educational level				
University degree	1.00	1.00	1.00	1.00
Complete secondary	1.18 (0.94–1.46)	1.02 (0.79–1.30)	1.03 (0.82–1.29)	1.07 (0.86–1.34)
Complete elementary	1.77 (1.01–3.11) *	0.93 (0.57–1.51)	0.94 (0.60–1.49)	0.82 (0.52–1.29)
Incomplete elementary	1.69 (0.93–3.08)	1.79 (1.09–2.94) *	1.74 (1.06–2.85) *	2.10 (1.25–3.52) †
Smoker				
Never	1.00	1.00	1.00	1.00
Past	1.26 (1.01–1.57) *	1.79 (1.42–2.26) ‡	0.93 (0.74–1.16)	1.15 (0.93–1.42)
Current	2.07 (1.46–2.93) ‡	2.53 (1.80–3.56) ‡	1.76 (1.27–2.43) ‡	1.87 (1.36–2.56) ‡
Physical activity				
Insufficient	1.00	1.00	1.00	1.00
Moderate	1.22 (0.95–1.56)	1.36 (1.05–1.76) *	0.86 (0.67–1.10)	0.95 (0.75–1.21)
Vigorous	1.03 (0.73–1.44)	1.32 (0.89–1.98)	1.27 (0.88–1.83)	1.29 (0.91–1.83)
Family history of CVD	1.37 (1.12–1.68) †	1.18 (0.95–1.47)	1.08 (0.88–1.33)	1.15 (0.94–1.40)
Body mass index (kg/m^2^)	1.05 (1.03–1.07) ‡	1.06 (1.04–1.09) ‡	1.04 (1.02–1.07) ‡	1.05 (1.03–1.07) ‡
Dyslipidemia	1.20 (0.98–1.46)	1.73 (1.40–2.14) ‡	1.29 (1.06–1.58) *	1.52 (1.25–1.83) ‡
Hypertension	1.52 (1.24–1.88) ‡	1.62 (1.30–2.03) ‡	2.09 (1.71–2.57) ‡	1.96 (1.61–2.39) ‡
Diabetes	1.07 (0.80–1.43)	1.35 (1.03–1.75) *	1.33 (1.03–1.72) *	1.46 (1.13–1.89) †

OR (95% CI), Odds ratio (95% confidence interval). * *p* ≤ 0.05; † *p* ≤ 0.01; ‡ *p* ≤ 0.001. AAC indicates aortic arch calcium; ATAC, ascending thoracic aortic calcium; CVD, cardiovascular disease; DTAC, descending thoracic aortic calcium; ADTAC, ascending plus descending thoracic aortic calcium.

## Data Availability

The data presented in this study are available on request from the corresponding author. The data are not publicly available due to ethical constraint.

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
