# Peer review of "Segmental Evaluation of Thoracic Aortic Calcium and Their Relations with Cardiovascular Risk Factors in the Brazilian Longitudinal Study of Adult Health (ELSA-Brasil)"

_cells, 2021, doi:10.3390/cells10051243_

Round 1
Reviewer 1 Report
Dear Authors, I must congratulate team for their impressive work in relation to aortic calcification.
Manuscript is interesting and discussed very important topic of aortic calcification in the different segments and its correlation for risk factors. Manuscript well written and only minimal English editing is required. However, publication in cells journal seems a bit off the from journal scope, as it looks like clinically related paper.
I agree in general with author idea that segmental calcification does correlate with patient clinical group. Few question and points of discussion to author:
- Author rightfully mention what common area of calcification is aortic arch. Why? What is the difference what makes arch so special? It should be discussed in the manuscript.
- I would like to check, what is clinical significance of this finding? Is the any direction author like to develop this lead? Correlation with some blood biological sample would be consider.
- It is important to mention that arch has specific hemodynamic characteristic leading to wall stress. It should be mention/discussed. (Role of Vascular Smooth Muscle Cell Plasticity and Interactions in Vessel Wall Inflammation by C.Shanahan. Frontiers in Immunology 11, 3053)
- Also, embryologically origin of ascending aorta and arch/descending aorta is different and it should be discussed in the light of difference in calcification (Genetic and Epigenetic Mechanisms Underlying Vascular Smooth Muscle Cell Phenotypic Modulation in Abdominal Aortic Aneurysm. R Gurung, et al. International Journal of Molecular Sciences 21 (17), 6334)
Author Response
Response to Reviewer 1 Comments
Dear Reviewer,
Thank you very much for your valuable comments and suggestions. Please find below our answers to your questions.
Point 1: Author rightfully mention what common area of calcification is aortic arch. Why? What is the difference what makes arch so special? It should be discussed in the manuscript.
Response 1: Thank you for making this point. Thoracic aortic calcium (TAC) has generally been analysed using the same tomographic scan obtained for CAC score, which does not include the aortic arch, therefore studies that evaluated the three aortic segments separately are very scarce. Craiem et al. 2014 demonstrated that aortic arch and the proximal descending aorta, hidden in standard TAC evaluations, concentrated most of the calcifications. In addition, these authors showed that middle-aged women were more prone to have calcifications in those hidden portions and, for this reason, they became candidates for reclassification of cardiovascular risk.
In order to address this point, we have added this discussion in line 659 to clarify the anatomical landmarks used when TAC is evaluated using CAC scans.
Point 2: I would like to check, what is clinical significance of this finding? Is the any direction author like to develop this lead? Correlation with some blood biological sample would be consider.
Response 2: In relation to the clinical significance, see the answer to question one. As far as we know, there is no correlation between calcium in aortic segments and biological blood samples used in routine clinical practices.
Point 3: It is important to mention that arch has specific hemodynamic characteristic leading to wall stress. It should be mention/discussed. (Role of Vascular Smooth Muscle Cell Plasticity and Interactions in Vessel Wall Inflammation by C.Shanahan. Frontiers in Immunology 11, 3053)
Response 3: Thank you for making this point. We have included the topic and the related references in the discussion section (line 681). Please, see below.
“The differences in the prevalence of calcium in each aortic segment may be explained in part by the effect of sheer forces on the wall, once the ascending aorta usually is larger, has high blood velocity and no branches [29]. Though high velocity can cause shear stress, most atherosclerotic lesions occur at sites where shear stresses are low but rapidly fluctuating, such as branch vessels or where there are abrupt changes in vessel diameter, both mostly occurring in the aortic arch followed by the descending aortic segment [30]. The hemodynamic conditions with the vasculature mean that vascular smooth muscle cells (VSMCs) are under constant mechanical stress. [31] These mechanical forces, such as shear stress, have been linked to premature cellular aging, including oxidative stress leading to vascular calcification. [31] Further, the VSMCs colonizing the aortic arch derive from cardiac neural crest cells, whereas those populating the descending aorta derive from the mesoderm [14]. The VSMCs are central players in vessel wall inflammation and evidence has shown that VSMCs impact every step of atherosclerosis. Studies have demonstrated that the inflammatory process may critically depend on VSMCs plasticity and their ability to switch between different phenotypes. [32] One of the reasons that smooth muscle cells identification and fate tracing is complex is related to the differences in the embryonic origin of VSMCs. [33]”
Point 4: Also, embryologically origin of ascending aorta and arch/descending aorta is different and it should be discussed in the light of difference in calcification (Genetic and Epigenetic Mechanisms Underlying Vascular Smooth Muscle Cell Phenotypic Modulation in Abdominal Aortic Aneurysm. R Gurung, et al. International Journal of Molecular Sciences 21 (17), 6334).
Response 4: Thank you again. Please, see the answer to point 3 above.

Reviewer 2 Report
In this study Jesiana Ferreira Pedrosa et al. have evaluated the presence of calcifications in the ascending, arch and descending aorta segments to investigate the associations of segmental TAC with different risk factors. They found that 62% of the patients had TAC in the aortic arch (AAC>0), a region that is usually invisible in routine CAC scans. The study has identified the risk factors associated with each segmental TAC presence. Finally, the authors claim that AAC could have an incremental predictive value beyond traditional risk factors to predict CVD risk.
The article is well written, the objectives are clear and the methodology is adequate. The topic is of interest for the medical community. Please, find below my comments and suggestions to improve your study.
- In Table 1 you incorporated a column for ascending+descending TAC (ADTAC>0). I assume that you wanted to compare what happens in routine CAC studies where the AAC is hidden. Why didn’t you add this column in table 2? Are ADTAC>0 risk factors different from AAC>0?
- In the results section the authors mention “Figure 2 demonstrates no sex-related differences in the prevalence of calcium in any aortic segments”. Are women and men of the same age? Why isn’t this comparison adjusted for age and BSA (or BMI)?
- Values and confidence intervals for the odds ratio in table 2 are somehow confusing to me. Are these values calculated per unit increase of the cofactor? For example, +1 year aging corresponds to an odd ratio of 1.1 similar to +1kg/m2 increase? Sometimes odd ratios are expressed per 1-SD increase to have a more clear view in continuous variables. Also, some 95% confidence intervals seem to have equal values. Please, clarify.
- Is the multivariate model adjusted for anti-HTA medication? What does it happen if you add this cofactor? Please comment.
- Usually TAC presence and extent are associated with body surface area since the size of the patient is correlated with the size of the aorta and the amount of calcium. Why did you use BMI? What happens if you exchange BMI for BSA in the multivariate analysis?
- It is surprising that you found a prevalence of 23% for ATAC when this segment is short (considering how you adopted the ascending aorta segment) and usually free of calcium. Did you measure the length of each aortic segment? Is this length associated with TAC presence? Did you evaluate the association of aortic diameter with TAC presence? (See DOI:10.1016/j.rec.2016.01.031).
- At the end of the first paragraph of the introduction the authos claim that “our findings suggest that extending the evaluation of arterial calcifications to AAC may bring different information regarding risk prediction, particularly among women.” Please, consider to rephrase this sentence because in my opinion there is no evidence in the manuscript to support this assertion.
Minor comments:
- Did you find calcifications in the ligamentum arteriosum? Did you include these calcifications in the arch or the descending aorta portions?
- The anatomical definition of the aortic segments does not agree with the portions adopted in your study. I understand that the aortic regions adopted in the study correspond with the hidden portion in routine CAC studies but maybe this discrepancy might be pointed out.
- Did you evaluate risk factors within patients with TAC, for example exploiting the TAC score value to quantify the extent of the calcifications? Is this a limitation of the study?
- Body mass index (BMI) was obtained by weight (Kg)/height SQUARED (m2).
- When you mention that a correlation study was performed with a random sample of 50 CT scans. Did you mean an inter- and intra-observer study? Instead of “by other general radiologist” maybe a second radiologist is better
Thank you for the opportunity to read and evaluate this manuscript.
Author Response
Dear Reviewer,
Thank you very much for your careful reading and valuable comments. We took all of them into consideration when preparing this revised version. Please, see the enclosed file.

Round 2
Reviewer 2 Report
I observe a noticeable improvement in the manuscript after this revised version. Please find below some minor comments:
Response 1: I do not understand when you mention “…except for family history of CVD, probably because of the prevalence of calcification is highest in the descending aorta”. In Table 1, neither DTAC nor ADTAC are significantly associated with family history of CVD.
Response 2: Reading your answer to Point 5, you may probably choose to adjust for BMI instead of BSA because BSA was not informed in your manuscript.
Response 3: I find better and more readable the table 2 with two decimal places. I saw that you also changed the number of decimal places in table 1. If possible, I would leave table 1 with only 1 decimal place.
Responses 5: Thank you for this BSA analysis. Since results in table 2 did not globally change and you do not use BSA along the manuscript, I would suggest removing supplementary table 2 from the final version.
Thank you for the opportunity to evaluate this manuscript.